# Projecting Lifetime Health Outcomes and Costs Associated with the Ambient Fine Particulate Matter Exposure among Adult Women in Korea

**DOI:** 10.3390/ijerph19052494

**Published:** 2022-02-22

**Authors:** Gyeyoung Choi, Yujeong Kim, Gyeongseon Shin, SeungJin Bae

**Affiliations:** 1College of Pharmacy, Ewha Womans University, Seoul 03760, Korea; 191ipg02@ewhain.net (G.C.); cmind96@naver.com (Y.K.); sunny628@g.ewha.ac.kr (G.S.); 2Korean Health Insurance Review & Assessment Service, Wonju 26465, Korea

**Keywords:** fine particulate matter, PM_2.5_, healthcare cost, QALYs, Markov model

## Abstract

We sought to estimate the lifetime healthcare costs and outcomes associated with the exposure to the escalated concentration of fine particulate matter (particle size < 2.5 μm, PM_2.5_) among adult Korean women. We adapted a previously developed Markov model, and a hypothetical cohort composed of Korean women was exposed to either a standard (15 μg/m^3^) or increased (25 μg/m^3^) concentration of PM_2.5_. The time horizon of the analysis was 60 years, and the cycle length was 1 year. The outcomes were presented as direct healthcare costs and quality-adjusted life years (QALYs), and costs were discounted annually at 5%. Deterministic and probabilistic sensitivity analyses were performed. The model estimated that when the exposure concentration was increased by 10 μg/m^3^, the lifetime healthcare cost increased by USD 9309, which is an 11.3% increase compared to the standard concentration group. Women exposed to a higher concentration of PM_2.5_ were predicted to live 30.64 QALYs, compared to 32.08 QALYs for women who were exposed to the standard concentration of PM_2.5_. The tendency of a higher cost and shorter QALYs at increased exposure was consistent across a broad range of sensitivity analyses. The negative impact of PM_2.5_ was higher on cost than on QALYs and accelerated as the exposure time increased, emphasizing the importance of early intervention.

## 1. Introduction

Air pollution represents one of the biggest environmental risks to health [1]. In 2012, more than three million deaths were attributable to ambient air pollution [2]. Among the pollutants, particulate matter with the diameter less than 2.5 μm (PM_2.5_) is known to be associated with the increased morbidity and mortality of various diseases [3]. PM_2.5_ penetrates within the respiratory tract and circulates in the blood stream due to its small size. As a result, PM_2.5_ affects not only the respiratory system but also the cardiovascular system and can cause various health problems. In fact, PM_2.5_ was the fifth leading cause of death worldwide following high blood pressure and smoking [4].

The World Health Organization (WHO) Air Quality Guidelines (AQGs) has recommended an annual average PM_2.5_ concentration of 10 μg/m^3^ as the target value and three interim targets (IT; IT-1 35 μg/m^3^, IT-2 25 μg/m^3^, IT-3 15 μg/m^3^), which have been shown to be achievable with successive and sustained abatement measures [5]. These concentrations were chosen based on the significance of their effect on survival, where the AQG target value was the lowest level at which the total, cardiopulmonary, and lung cancer mortality were shown to increase with more than 95% confidence in response to PM_2.5_ in the American Cancer Society study. In South Korea, the PM_2.5_ level has been continually maintained over the guideline value since 2015 when the South Korean government started the official observation of PM_2.5_ [6]. However, due to geographical and seasonal reasons, the airborne fine particulate matters from foreign high-emission areas add to the burden of domestic pollution [7]. Therefore, despite the efforts of the government to manage the annual concentration, the potential negative health effects of PM_2.5_ have been of great concern among the Korean people.

Though there have been many studies reporting an increased disease risk associated with the elevated PM_2.5_ [8,9,10,11], studies projecting the lifetime economic effect of diseases due to PM_2.5_ are limited. In an economic burden of disease study performed in 111 cities, the total economic cost caused by particulate matter pollution in 2004 was estimated to be approximately USD 29,178.7 million [12]. If the concentration of PM_2.5_ decreases by 10 μg/m^3^, more than USD 22 million of economic benefit will occur annually in Seoul, South Korea [13]. Moreover, life expectancy will be lengthened by 0.35 years [14]. However, there have been no studies estimating the lifetime cost and a much lower quality of life, which is reported to be significantly affected by PM_2.5_ exposure [15,16]. The purpose of this study was to estimate the economic and health outcomes of ambient PM_2.5_ exposure for a lifetime among Korean women.

## 2. Materials and Methods

### 2.1. Markov Model

We have adopted a previously developed Markov state-transition model to simulate the natural history of PM_2.5_ exposure (Figure 1) [17]. At the start of the model, the cohort was exposed to either standard or increased concentration of PM_2.5_. Subsequent Markov model pathways were same with four diseases and their corresponding health states. The whole cohort started from event-free health state and transferred to each health state according to the probabilities derived from reference data. All health states were mutually exclusive and collectively exhaustive so the patients could be assigned to only one health state at any given time [18,19].

The four diseases, lung cancer, myocardial infarction, stroke, and COPD, were selected based on previous systematic reviews and official reports, in which those diseases were named as some of the most affected diseases due to PM_2.5_ exposure [1,20,21,22,23]. Of these four diseases, myocardial infarction, stroke, and COPD were further sorted by progression period (e.g., first year and following years) because transition probabilities, quality of life, and treatment cost vary significantly [24,25,26]. To investigate the long-term effect of PM_2.5_ exposure, the analysis period was 60 years, and the cycle length was 1 year. The outcomes were presented as US dollars (USD) and QALYs, and the costs were discounted at 5% annually to reflect people’s positive time preference [27,28]. TreeAge Pro^®^ 2020 software was used to build the simulation model.

### 2.2. Target Population

A hypothetical cohort of 10,000 Korean women aged 40 years old was analyzed. We targeted middle-aged women because many previous studies on PM_2.5_ have been performed targeting middle-aged female population [29,30]. To assess the cost and health risk of increased exposure of PM_2.5_, the study population was assumed to be consistently exposed to either increased or standard concentration of PM_2.5_. Increased concentration was defined as 25 μg/m^3^, which is the average annual PM_2.5_ concentration of South Korea in 2017 [6]. Standard concentration was set to 15 μg/m^3^, which is the interim target 3 (IT-3) concentration of PM_2.5_ established by WHO.

### 2.3. Input Data

A systematic review was performed to obtain the increased risk of disease incidence due to PM_2.5_ exposure (Appendix A) [31]. Firstly, the search terms and PICO (population, intervention, comparison, outcomes) were set, and then we searched through PubMed and conducted an additional search in Google Scholar. The search strategy is shown in Table 1. After the search of electronic database, detailed criteria such as the age of the population or exposure concentration to PM_2.5_ were checked by two authors (G.C, Y.K) Any disagreement between the two authors over the eligibility of studies was resolved through discussion with a third author (G.S). Data extraction included sample size of study, age, gender, PM_2.5_ exposure status, PM_2.5_ concentration increment, and outcomes (relative risk). Cohort studies with large sample size were preferred and final selection was based on the similarity of the study cohort to our target population.

Only direct medical costs were included, and non-medical costs such as transportation cost or lost productivity cost were excluded. Domestic studies were preferentially searched since the treatment costs vary by country. Each cost was adjusted by the medical care component of the Consumer Price Index (CPI) in Korea using the equation below [32]. The adjusted costs were then transferred to 2020 US dollars [33].
$$Costs_{Current\;year}=Costs_{Base\;year}\times \frac{CPI_{Current\;year}}{CPI_{Base\;year}}$$

QALY was chosen as a tool to quantify the impact of PM_2.5_ exposure on health-related quality of life. The utilities of respective health states were obtained through literature search. The baseline utilities of age and sex-specific Korean general population were sourced from Korea National Health and Nutrition Examination Survey [34]. The utilities of event-free women of standard and increased exposure group were assumed to be the same, which is a conservative assumption.

### 2.4. Sensitivity Analysis

Univariate and probabilistic sensitivity analyses were performed to investigate the robustness of the model because our study was based on several assumptions. Univariate sensitivity analysis was conducted on discount rate (0%, 3%, 7%), time horizon (5, 10, 20, 40 years), and relative risks (95% confidence interval). For the probabilistic sensitivity analysis (PSA), 10,000 times of second-order Monte Carlo simulations were conducted on the relative risks, utilities, and costs. We applied a lognormal distribution for relative risks, a beta distribution for utilities, and a gamma distribution for costs, with the reference of previous studies. The applied distribution for each variable is presented in Table 2. The PSA result was visualized by a scatterplot.

## 3. Results

### 3.1. Input Data

For the relative risks, eight studies were selected through systematic review [29,30,35,36,37,38,39,40]. Relative risk data from the eight studies applied in our model and each reference are shown in Table 3. Among the eight studies, two of them were meta-analysis studies and six were cohort studies. The study populations were from the US, Canada, Europe, South America, and Taiwan. The sizes of the study cohorts were from 65,893 to 367,383 and the follow-up period was from 6 to 14 years. The incremental PM_2.5_ concentration was 10 μg/m^3^ in five studies and 5 μg/m^3^ in three studies. Because the data that matched the characteristics of our target population were not available from domestic studies, relative risks were sourced from international studies. Incidence and mortality rates for each disease were sourced from Korea Statistics. The annual incidence rates, mortality rates, and relative risks used in the model are summarized in Table 3.

Annual costs and QALY data applied for the model and their references are summarized in Table 4. The costs for each health states were referred from domestic studies including cost data estimated from the Korean National Health Insurance database. The age and sex-specific EQ-5D of the general Korean population was sourced from Korea National Health and Nutrition examination survey (2015) [34].

### 3.2. Base-Case Analysis

The model estimated that the increased exposure to PM_2.5_ would cost USD 9309 per woman for lifetime healthcare, whereas the lifetime healthcare cost would be USD 8367 per woman when exposed to the standard PM_2.5_ concentration. The predicted QALYs were 32.08 and 30.64 for increased exposure and standard exposure, respectively (Figure 2). The lifetime healthcare cost increased by 11.3% and QALYs decreased by 4.5% in the case of increased exposure to PM_2.5_.

### 3.3. One-Way Sensitivity Analysis

The results of one-way sensitivity analysis are summarized in Table 5. The one-way sensitivity analysis demonstrated that exposure to an increased concentration of PM_2.5_ generally shows higher healthcare costs and lower QALYs compared to the standard exposure group across various assumptions. Specifically, when the relative risk of lung cancer incidence was varied, the negative impact of the increased exposure to PM_2.5_ was the highest resulting in a 23% increase in lifetime healthcare cost. When the discount rate was changed by 0%, 3%, and 7%, the costs were USD 38,589, USD 15,753, and USD 5800 at the increased exposure, which were 7.2%, 9.8%, and 12.6% increases compared to the cost of standard exposure, respectively. As the time horizon increased, the direct healthcare costs of PM_2.5_ exposure escalated from USD 189 for 5 years to USD 8254 for 40 years. This indicates that the negative economic impact associated with PM_2.5_ exposure increased over time, and Figure 2 also suggests that the negative impact accelerates as time progresses.

### 3.4. Probabilistic Sensitivity Analysis

Probabilistic sensitivity analysis showed that the costs of the increased concentration group varied from USD 5570 to USD 14,045 (134.6%), while the costs of the standard concentration group varied from USD 5382 to USD 12,628 (152.1%). However, the variation for QALYs was smaller (81.7% vs. 88.4%) between the two groups (Table 6). The result of the probabilistic sensitivity analysis is visualized in Figure 3.

### 3.5. Model Validation

External validation was performed to compare our result to actual observed epidemiological mortality data. In our study, we compared the lung cancer mortality projection result with the observed data reported by Li et al. [21]. The study involved a cohort of 118,551 final participants, 58.9% of which were women, and the follow-up period was 15 years. The cohort was exposed to 31–54 μg/m^3^ of PM_2.5_. For direct comparison, the analysis period of our Markov model was set to 15 cycles. Li and colleagues reported 77.34 lung cancer deaths per 100,000 persons per year for a PM_2.5_ exposed condition, which was higher than our study projection (56.07 lung cancer deaths per 100,000 persons per year (Table 7)) yet understandable, given the difference in the PM_2.5_ exposure.

## 4. Discussion

Our model estimated that adult Korean women exposed to an increased concentration of PM_2.5_ incurred an additional USD 942 in their lifetime and lived 1.44 QALYs shorter compared to the standard exposure group. The one-way sensitivity analysis showed that higher healthcare costs and shorter QALYs were expected for the increased exposure group, regardless of various assumptions. For model validation, the results were compared with the external literature, which studied the relative risk of lung cancer death due to PM_2.5_ exposure. The predicted mortality rate from our model was 0.005607, which was slightly lower than the observed mortality rate of 0.007734 reported by Li et al. (2020) [21]. However, this difference can be explained by the fact that the mortality rate due to lung cancer in China is higher than in South Korea [49], and the study population of Li et al. (2020) was exposed to 31–54 μg/m^3^ of PM_2.5,_ which is higher than 25 μg/m^3^, the exposure concentration of the hypothetical cohort in our model. For further validation, the incidence rate of stroke was compared to that of another study [50] where the observed incidence rate was 393 cases per 100,000 person years (0.03930). Our estimation (0.03011) was comparable to the value obtained from the observation data. Therefore, we concluded that our model is valid, and the result of our model is acceptable. Though the data are not shown, we estimated life year expectancy of each cohort by rewarding each cycle being 1 without adjusting the quality of life. When the disease burden was not considered, the expected life years of 40-year-old Korean women were 39.47 years and 41.34 years for increased exposure and standard exposure, respectively. The life expectancy of 40-year-old Korean women was 47.3 years in 2020, based on the lifetable reported by Statistics Korea [51]. This shows that our study provides a conservative estimate.

The economic loss and health impact due to ambient particulate matters has been reported in several previous studies, yet our study is the first attempt to project the long-term effect by using a simulation model. While most studies reported the PM_2.5_-induced economic loss as a regional unit [12], Yin et al. (2017) reported that the PM_2.5_ concentration in Beijing (40.26–92.30 μg/m^3^) induces an economic loss of USD 18 to 147 per capita yearly [52]. These data were calculated by the Willingness to Pay (WTP) or Amended Human Capital (AHC) method, and they include the disutility of illness, productivity loss, medical expenditures associated with illnesses, and expenditures on disease prevention. Because the projection method used in our study and Yin et al.’s (2017) study is different in nature, it is not appropriate to directly compare the results between the two studies. The relatively low health cost in our study is due to not only the analysis method or exposure concentration difference but also the conservative assumptions defined in our model. In our model, we included four diseases in circulatory, respiratory, and neoplasm (lung cancer), which were known to be highly related to PM_2.5_ exposure. However, Yin et al.’s (2017) study included additional endocrine/nutritional/metabolic diseases, mental and behavioral disorders, and nervous system diseases and this could increase the cost.

The Markov model method was used in our study to extrapolate the lifetime effect of PM_2.5_ based on the data adopted from the existing literature, such as transition probabilities between health states. However, because of this, there are some methodological limitations in our study. First, the relative risks for the diseases used in our model were derived from international studies. The systematic review by Lim et al. (2020) reported the hazard ratios for mortalities due to PM_2.5_ increase in the Korean population [53]. However, this study was conducted targeting only the elderly population and could not represent the mortality of middle-aged Korean women. Kim et al. (2018) studied 570 thousand deaths across three metropolitan cities in Korea and reported that PM_2.5_ is significantly associated with daily mortality of all causes, and respiratory and cardiovascular diseases [54]. According to the study, the estimation can be updated when the relative risks of the domestic population suitable for our model is reported. Secondly, the effect of PM_2.5_ on the economic cost and health outcomes may have been underestimated since only four diseases were included in the model. There is gaining evidence in the relationship between fine particulate matter exposure and various diseases. Some studies have reported that exposure to PM_2.5_ is related to the increased morbidity of asthma attacks, diabetes, obesity, Alzheimer’s, Parkinson’s, dementia, mild cognitive disorders, and bladder cancer [52,55], yet the clinical relevance is inconsistent [1] and further study is needed. In addition, we assumed that the relative risks were constant regardless of the exposure period, which is a clear limitation. However, the effect of PM_2.5_ is likely to accumulate for prolonged exposure [56,57]. Finally, although we focused on the effect of PM_2.5_ in this study, there are various environmental factors that we did not put into the model, such as toxic elements and possible medicinal interference during the cycles, since the quantified impact (such as relative risks) was either not available or not statistically significant [58,59]. Despite those limitations, our study is the first attempt to project the economic and quality of life impact of PM_2.5_ exposure based on a simulation model, which could eliminate the effect of variables other than the exposure to PM2.5 itself. 

## 5. Conclusions

The negative impact of PM_2.5_ was higher on the healthcare costs than on the QALYs, and accelerated as the exposure time accumulated. The results were consistent across various assumptions. A prompt, aggressive intervention is needed to reduce burdens associated with PM_2.5_ exposure.

## Figures and Tables

**Figure 1 ijerph-19-02494-f001:**
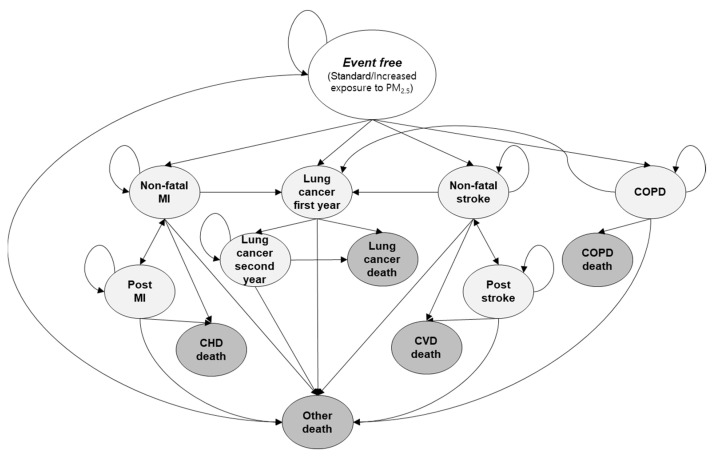
Health states and disease progression for Korean adult women who are exposed to ambient PM_2.5_. CHD, coronary heart disease; CVD, cardiovascular disease; MI, myocardial infarction.

**Figure 2 ijerph-19-02494-f002:**
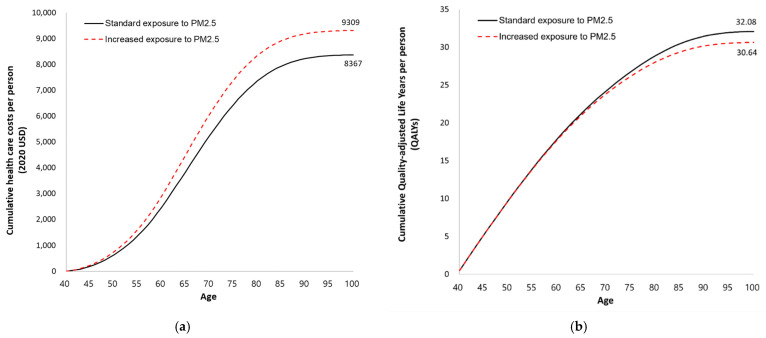
Cumulative lifetime healthcare costs and quality-adjusted life years for Korean adult women who were exposed to either increased or standard concentration of PM_2.5_. (**a**) Lifetime healthcare costs; (**b**) quality-adjusted life years.

**Figure 3 ijerph-19-02494-f003:**
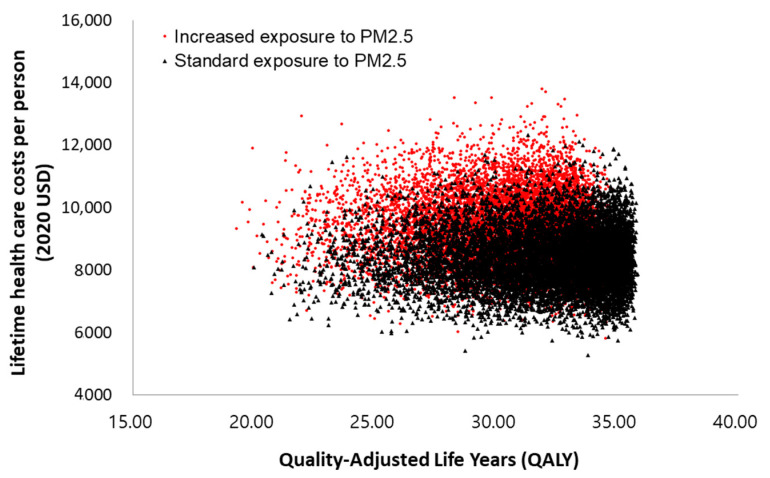
Scatter plot of the probabilistic sensitivity analyses results.

**Table 1 ijerph-19-02494-t001:** Search strategy for the systematic review of relative risks of diseases used in the model.

Step	Search Strategy
#1	woman or female
#2	particulate matter or PM_2.5_
Lung Cancer	
#3	lung cancer or lung carcinoma
#4	#1 and #2 and #3
#5	#1 and #2 and #3 and (relative risk or hazard ratio) and (incidence rate or prevalence or mortality)
#6	Filters: English, Korean, Adult: 19+ years
Myocardial Infarction	
#3	myocardial infarction or cardiovascular disease or ischemic heart disease or coronary heart disease
#4	#1 and #2 and #3
#5	#1 and #2 and #3 and (relative risk or hazard ratio) and (incidence rate or prevalence or mortality)
#6	Filters: English, Korean, Adult: 19+ years
Stroke	
#3	stroke or cerebrovascular disease or cerebral hemorrhage or cerebral infarction
#4	#1 and #2 and #3
#5	#1 and #2 and #3 and (relative risk or hazard ratio) and (incidence rate or prevalence or mortality)
#6	Filters: English, Korean, Adult: 19+ years
COPD	
#3	chronic obstructive pulmonary disease or COPD
#4	#1 and #2 and #3
#5	#1 and #2 and #3 and (relative risk or hazard ratio) and (incidence rate or prevalence or mortality)
#6	Filters: English, Korean, Adult: 19+ years

COPD, chronic obstructive pulmonary disease.

**Table 2 ijerph-19-02494-t002:** Distribution of variables for probabilistic sensitivity analysis.

Variables	Distribution
**Relative risks**	
Lung cancer incidence	Lognormal
Lung cancer mortality	Lognormal
MI incidence	Lognormal
MI mortality	Lognormal
Stroke incidence	Lognormal
Stroke mortality	Lognormal
COPD incidence	Lognormal
COPD mortality	Lognormal
**Utilities**	
Lung cancer, first year	Beta
Lung cancer, second year	Beta
MI	Beta
Post MI	Beta
Stroke	Beta
Post stroke	Beta
COPD	Beta
**Health care costs**	
Lung cancer, first year	Gamma
Lung cancer, second year	Gamma
Lung cancer death	Gamma
Non-fatal MI	Gamma
Post MI	Gamma
CHD death	Gamma
Non-fatal stroke	Gamma
Post stroke	Gamma
CVD death	Gamma
COPD	Gamma
COPD death	Gamma

CHD, coronary heart disease; CVD, cardiovascular disease; MI, myocardial infarction; COPD, chronic obstructive pulmonary disease.

**Table 3 ijerph-19-02494-t003:** Annual incidence and mortality rates for each disease states and the relative risks related to PM_2.5_ exposure used in the model.

Disease		Age		Ref	Relative Risk	Ref
Lung cancer	Incidence rate	40–49 50–59 60–69 70–	0.0001 0.0003 0.0007 0.0014	[34]	1.42 (1.02–1.98)	[37]
	Mortality rate	-	0.2109	[41]	1.27 (1.03–1.56)	[40]
Myocardial infarction	Incidence rate	45 55 65 75 85	0.0004 0.0013 0.0033 0.006 0.0085	[42]	1.22 (1.04–1.44)	[36]
	Mortality rate	45 55 65 75 85	0.0168 0.0324 0.0618 0.1152 0.2076	[42]	1.20 (1.02–1.41)	[29]
Stroke	Incidence rate	45 55 65 75 85	0.0011 0.0029 0.0076 0.0158 0.025	[42]	1.28 (1.02–1.61)	[30]
	Mortality rate	45 55 65 75 85	0.0046 0.0112 0.0263 0.0604 0.1295	[42]	1.34 (0.94–1.91)	[35]
COPD	Incidence rate	40–49 50–59 60–69 70–	0.008 0.024 0.114 0.136	[34]	1.08 (1.04–1.11)	[38]
	Mortality rate	75 85	0.0002 0.0009	[41]	1.169 (1.136–1.203)	[39]

COPD, chronic obstructive pulmonary disease; Ref, reference.

**Table 4 ijerph-19-02494-t004:** Annual costs (per person) and utility used in the model.

State	Cost, Year 2020 (USD)	Ref	Utility	Ref
Lung cancer, first year	19,495	[25]	0.61	[43]
Lung cancer, second year	6180	[25]	0.50	[43]
Lung cancer death	17,089	[44]	-	
Non-fatal MI	7026	[45]	0.71	[46]
Post MI	1156	[45]	0.75	[46]
CHD death	1494	[45]	-	
Non-fatal stroke	7260	[45]	0.63	[47]
Post stroke	941	[45]	0.72	[47]
CVD death	2062	[45]	-	
COPD	809	[24]	0.8	[48]
COPD death	2577	[24]	-	

CHD, coronary heart disease; CVD, cardiovascular disease; MI, myocardial infarction; COPD, chronic obstructive pulmonary disease; Ref, reference.

**Table 5 ijerph-19-02494-t005:** One-way sensitivity analyses for Korean adult women who are exposed to increased concentration of ambient PM2.5 compared with women exposed to standard concentration.

Parameters	PM_2.5_ Exposure	Cost (USD)	QALYs	Incremental Cost (USD)	Difference (%)	Incremental QALYs	Difference (%)
Discount rate (%)							
0	Standard	36,013	32.08				
	Increased	38,589	30.64	2575	7.2%	−1.44	−4.5%
3	Standard	14,353	-				
	Increased	15,753	-	1400	9.8%	−0.51	−2.7%
5	Standard	8367	-				
	Increased	9309	-	942	11.3%	−0.28	−1.9%
7	Standard	5152	-				
	Increased	5800	-	648	12.6%	−0.16	−1.4%
Time horizon (years)							
5	Standard	158	4.75				
	Increased	189	4.74	31	20.0%	0.00	−0.1%
10	Standard	573	9.31				
	Increased	679	9.29	106	18.5%	−0.02	−0.2%
20	Standard	2349	17.55				
	Increased	2751	17.43	402	17.1%	−0.12	−0.7%
40	Standard	7274	28.70				
	Increased	8254	27.89	980	13.5%	−0.81	−2.8%
Relative risk for Lung Cancer incidence							
Lower bound of 95% CI	Standard	8367	32.08				
	Increased	8515	31.57	148	1.76%	−0.52	−1.61%
Upper bound of 95% CI	Standard	8367	32.08				
	Increased	10,264	29.52	1897	22.68%	−2.56	−7.97%
Relative risk for Lung Cancer mortality							
Lower bound of 95% CI	Standard	8367	32.08				
	Increased	9553	30.77	1186	14.18%	−1.31	−4.09%
Upper bound of 95% CI	Standard	8367	32.08				
	Increased	9093	30.53	726	8.68%	−1.55	−4.84%
Relative risk for Myocardial Infarction incidence							
Lower bound of 95% CI	Standard	8367	32.08				
	Increased	9287	30.66	920	11.00%	−1.42	−4.42%
Upper bound of 95% CI	Standard	8367	32.08				
	Increased	9338	30.62	971	11.60%	−1.47	−4.57%
Relative risk for Myocardial Infarction mortality							
Lower bound of 95% CI	Standard	8367	32.08				
	Increased	9318	30.67	951	11.37%	−1.41	−4.41%
Upper bound of 95% CI	Standard	8367	32.08				
	Increased	9301	30.62	934	11.16%	−1.46	−4.56%
Relative risk for Stroke incidence							
Lower bound of 95% CI	Standard	8367	32.08				
	Increased	9240	30.71	873	10.43%	−1.37	−4.28%
Upper bound of 95% CI	Standard	8367	32.08				
	Increased	9405	30.56	1038	12.40%	−1.52	−4.75%
Relative risk for Stroke mortality							
Lower bound of 95% CI	Standard	8367	32.08				
	Increased	9337	30.74	970	11.60%	−1.34	−4.17%
Upper bound of 95% CI	Standard	8367	32.08				
	Increased	9272	30.51	905	10.82%	−1.57	−4.89%
Relative risk for COPD incidence							
Lower bound of 95% CI	Standard	8367	32.08				
	Increased	9193	30.71	826	9.88%	−1.37	−4.27%
Upper bound of 95% CI	Standard	8367	32.08				
	Increased	9393	30.59	1026	12.27%	−1.49	−4.64%
Relative risk for COPD mortality							
Lower bound of 95% CI	Standard	8367	32.08				
	Increased	9309	30.64	942	11.26%	−1.44	−4.48%
Upper bound of 95% CI	Standard	8367	32.08				
	Increased	9310	30.64	943	11.27%	−1.44	−4.49%

QALYs, quality-adjusted life years; COPD, chronic obstructive pulmonary disease.

**Table 6 ijerph-19-02494-t006:** Summary of the probabilistic sensitivity analyses results.

Statistic	Costs (USD)		QALYs	
	Increased Exposure to PM_2.5_	Standard Exposure to PM_2.5_	Increased Exposure to PM_2.5_	Standard Exposure to PM_2.5_
Mean	9352	8367	30.59	32.05
Std Deviation	1064	928	2.74	2.89
Minimum	5570	5382	18.78	19.75
2.50%	7410	6682	23.91	24.96
10%	8020	7201	26.65	27.83
Median	9304	8322	31.20	32.79
90%	10,743	9587	33.54	35.07
97.50%	11,583	10,316	34.20	35.46
Maximum	14,045	12,628	35.39	35.88

QALYs, quality-adjusted life years.

**Table 7 ijerph-19-02494-t007:** Result of the model validation analysis.

	Lung Cancer Mortality (Case/Person per Year)	
	Li et al. (2020)	Model
Increased exposure to PM_2.5_	0.007734	0.005607

## Data Availability

The data presented in this study are available in the Appendix A.

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
