# Peer review of "Projecting Lifetime Health Outcomes and Costs Associated with the Ambient Fine Particulate Matter Exposure among Adult Women in Korea"

_ijerph, 2022, doi:10.3390/ijerph19052494_

Round 1

Reviewer 1 Report

The assessment on the input data quality of the model is very less. For example, how about the assumptions from the authors systematic review? How much the model bias from the systematic review data?

Finally, how do the authors explain the model is good enough to support you conclusion?

Reviewer 2 Report

Comments to ijerph-1534949:

General comments:

The study determined the association between PM2.5 exposure to adult women and outcome and health costs based on a hypothetical cohort using a previously developed Markov state-transition model. It is interesting and important for the understanding of the impacts of PM2.5 pollution in ambient air on human health and costs. There are still some issues should be addressed before the publication. Some detailed information of the methods should be specified (see the specific comments bellow). The language should be further polished with English native speakers at the revision stage.

Specific comments:

  1. What is the principle or basis of the cohort hypothesis?And explain the reason why used the hypothetical cohort?
  2. In Table 1: the article did not find out how to screen the 40-year-old target population from the search results.
  3. Line129: No description was found for the eight studies presented in the article, which eight studies were specifically selected?
  4. Line 130-132: does the mismatch between the relatives risk and the disease data in the target population affect the model results?
  5. In Table 7: It is suggested to add some additional literature data in Section 3.5 and the first paragraph of the Discussion to strengthen the comparison.
  6. Line 198-201. how to get the expected life-years of 40-year-old Korean women were 41.43 years and 39.47 years for increased exposure and standard exposure through the additional analysis?The process or explanation of the analysis should be addressed
  7. Line 210-212. The method from Yin et al. is quite different from that of this study, suggest to deleting it.

Reviewer 3 Report

Dear Authors,

I have the following comments to your manuscript:

To your " Markov model " are missing references please include it.

Page 1, line 29

" PM2.5 penetrates within the respiratory tract and circulates in the blood stream due to its small size. As a result, PM2.5 affects not only the respiratory system but also cardiovascular system and can cause various health problems "

You have to consider also bigger as 2.5 micrometer size as well as smaller particles, because in reality the PM emissions from industries and vehicles are in different sizes.

For humans:

Big particles are filtered in human nose and in nasal sinus.

Smaller particles can penetrate into the respiratory tract/system into the lungs.

Even smaller particles can be filtered via alveolus.

Not all cardiovascular diseases are necessary caused by particles.

Important issues are also toxic gases, fats stacked in inner parts of the blood vessel and in artery. Combinations of abnormal obesity, occupation hazardous toxins and relatively high exposed time to increased PM concentration are crucial.

In the future it would be important to your model include the information about the toxic information PM, gas, smoking, overweight, fats, obesity etc.

Page 3, Figure 1

Enlarge the figure size, or enlarge the font size of the text in the figure

Page 6, table 4.

" Annual costs and utility used in the model. "

this information is per person per year or ?

Page 7, Figure 2

From this graph it is important in your model consider not only the age of the person but also the number of years exposed to the PM due to the hazardous occupation, or residence in locations with high PM exposure.

Page 10, line 217

" mental and behavioural disorder, and nervous system diseases"

these can be linked also to different issues like toxic gases, toxic elements, occupation with high hazardous exposure to toxic elements, overdose with different drug, narcotic, or combination of different medication etc.

therefore into the model these effect has to be also considered

Page 11, line 240 in the Conclusion

please rewrite this chapter again and include:

shortly explain what was your research about, topic

summarise what was your research goal/s

summarise your main finding and results.

At the end of the manuscript include list of all abbreviations mentioned in the text.

Author Response

Please see the attachment. Thank you for the comments.
